# Dynamics of Photoinduced Charge Carriers in Metal-Halide Perovskites

**DOI:** 10.3390/nano14211742

**Published:** 2024-10-30

**Authors:** András Bojtor, Dávid Krisztián, Ferenc Korsós, Sándor Kollarics, Gábor Paráda, Márton Kollár, Endre Horváth, Xavier Mettan, Bence G. Márkus, László Forró, Ferenc Simon

**Affiliations:** 1Department of Physics, Institute of Physics, Budapest University of Technology and Economics, Műegyetem Rkp. 3., H-1111 Budapest, Hungarybmarkus@nd.edu (B.G.M.); 2Semilab Co., Ltd., Prielle Kornélia U. 2, H-1117 Budapest, Hungary; ferenc.korsos@semilab.hu (F.K.);; 3Institute for Solid State Physics and Optics, HUN-REN Wigner Research Centre for Physics, P.O. Box 49, H-1525 Budapest, Hungary; 4KEP Innovation Center, Ch. du Pré-Fleuri 5, 1228 Plan-les-Ouates, Switzerland; kollar7@gmail.com (M.K.);; 5Stavropoulos Center for Complex Quantum Matter, Department of Physics and Astronomy, University of Notre Dame, Notre Dame, IN 46556, USA; 6ELKH-BME Condensed Matter Research Group, Budapest University of Technology and Economics, Műegyetem Rkp. 3., H-1111 Budapest, Hungary

**Keywords:** perovskites, photoconductivity, photonic applications, charge-carrier lifetime

## Abstract

The measurement and description of the charge-carrier lifetime ($\tau_{c}$) is crucial for the wide-ranging applications of lead-halide perovskites. We present time-resolved microwave-detected photoconductivity decay (TRMCD) measurements and a detailed analysis of the possible recombination mechanisms including trap-assisted, radiative, and Auger recombination. We prove that performing injection-dependent measurement is crucial in identifying the recombination mechanism. We present temperature and injection level dependent measurements in CsPbBr_3_, which is the most common inorganic lead-halide perovskite. In this material, we observe the dominance of charge-carrier trapping, which results in ultra-long charge-carrier lifetimes. Although charge trapping can limit the effectiveness of materials in photovoltaic applications, it also offers significant advantages for various alternative uses, including delayed and persistent photodetection, charge-trap memory, afterglow light-emitting diodes, quantum information storage, and photocatalytic activity.

## 1. Introduction

Metal-halide perovskites have garnered significant attention across various applications due to their low fabrication costs, ease of growth compared to silicon, and less sensitivity to crystalline imperfectness, while offering exceptional photonic properties [1,2,3]. These materials follow the ABX_3_ structure, where A can be an organic or inorganic constituent (such as CH_3_NH_3_ or Cs), B is typically Pb or Sn, and X is a halogen like I, Br, or Cl. This structural flexibility leads to a diverse array of compounds, enabling precise tuning of properties such as the optical band gap, mobility, crystal structure, and even isotopic content for nuclear conversion processes. By adjusting the halide content, the optical characteristics can be finely controlled [4,5]. Metal-halide perovskite solar cells, for instance, achieve remarkable photovoltaic efficiencies exceeding 26%, presenting a promising alternative to silicon-based cells [6]. Beyond solar cells, they are also utilized in photodetectors [7], X-ray [8], gamma-ray [9], neutron detectors [10], as well as gas sensors [11], with potential applications in harsh environments, including outer space [12,13].

Organic-hybrid perovskites, like CH_3_NH_3_PbX_3_ compounds, are widely studied [1,14,15,16,17] due to the added flexibility that the organic component provides in terms of charge transfer, crystalline structure, and symmetry [18]. This flexibility also influences factors like atom migration, stoichiometric imbalance, and enables the use of various spectroscopic techniques, including infrared [19] and magnetic resonance spectroscopy [20]. A significant drawback of organic-containing perovskites is their increased vulnerability to ambient factors like oxygen and humidity [21]. In contrast, inorganic perovskites demonstrate greater stability under ambient conditions, making them more suitable for applications.

CsPbBr_3_ emerged as a particularly appealing material among inorganic lead-halide perovskites [22]. Its attractiveness stems from several factors: a direct optical band gap centered within the visible spectrum, low susceptibility to moisture and air degradation [23], the use of Cs as a crucial element in scintillation detectors [24], and notable structural stability [25]. Thus, CsPbBr_3_ shows great potential for applications in solar cells [26], LEDs [4], lasers [27,28], photodetectors [7,29], and radiation detectors [30].

For most applications, the lifetime of photo-generated charge carriers ($\tau_{c}$) is a crucial parameter, as it directly influences photovoltaic and light-emission efficiency. Time-resolved photoluminescence (TRPL) in CsPbBr_3_ has been extensively studied [31,32,33,34,35]. This method primarily captures the radiative lifetime associated with band-to-band recombination. However, time-resolved photoconductivity measurements also account for non-radiative (i.e., non-photon-emitting) processes.

The latter includes the impurity assisted recombination (also known as Shockley–Read–Hall) and the Auger process. Time-resolved photoconductivity measurements in CsPbBr_3_ can be carried out using either a DC technique [30] (where the DC photoconductivity is detected under light irradiation) or microwave-detected photoconductivity [36,37]. The DC method is limited in time resolution to a few milliseconds, whereas the microwave technique can measure $\tau_{c}$ down to 100 nanoseconds [38,39]. Time-resolved microwave conductivity measurements have been performed on perovskite samples before [40,41,42] but on a limited temperature range.

We recently reported temperature and injection dependent measurements in the fully inorganic lead-halide perovskite CsPbBr_3_ (Ref. [43]). Here, we provide additional experimental data on further samples which essentially reproduce the previous observations. We also give a thorough theoretical modeling of the observed charge-carrier recombination mechanism. We discuss the possible alternatives of the charge carrier recombination processes including radiative, Auger, and Shockley–Read–Hall mechanisms. This analysis may also serve as a reference for future time-resolved photoconductivity measurements and we argue for the utility of injection dependent TRMCD studies. We observe from the detailed modeling that charge-carrier trap states dominate the charge-carrier lifetime. Variation of a few physical parameters allows to qualitatively account for the observed features in the whole temperature range.

## 2. Methods

### 2.1. Sample Preparation

The precursor solution of CsPbBr_3_ was prepared by dissolving PbBr_2_ and CsBr in a 2:1 molar ratio in dimethyl sulfoxide (DMSO). PbBr_2_ (98+%) and DMSO (99.8+%) were sourced from Thermo Scientific, CsBr (99%) from Alfa Aesar, and dimethylformamide (DMF, 99%) from Fisher Scientific (Waltham, MA, USA). The precursor solution was heated to 80 °C to accelerate the dissolution of the salts. After 10 h of stirring, a transparent solution was obtained, which was then filtered. This preparation method is a modified version of the technique reported by Dirin et al. [44].

The crystals are grown using the inverse temperature crystallization (ITC) technique, initially proposed by Saidaminov et al. [45,46]. In this method, the solubility of the perovskite decreases as the temperature is increased from 80 °C to 120 °C in DMSO. After the initial growth, the crystals are collected and used as seed crystals for further growth. These seed crystals are placed in a fresh precursor solution, which is gradually heated to 110 °C. The resulting single crystal is then harvested, rinsed with dimethylformamide (DMF), and dried.

### 2.2. The Experimental Setup

Figure 1a shows the block diagram of the instrument developed to measure photoconductivity in the perovskite materials. The sample is placed on the cold finger of a cryostat (Janis Inc., Woburn, MA, USA), and thus, we can detect the recombination process of charge carriers in the 10–300 K temperature range. Time-resolved detection of the photoconductivity traces is performed with a high-speed oscilloscope (Tektronix MDO3024, Beaverton, OR, USA), which is triggered by the Q-switch laser pulses, provided by a photodiode (DET36A/M, Thorlabs, Newton, NJ, USA) behind a beam sampler. A pulsed laser (NL201-2.5k-SH-mot, Ekspla, Vilnius, Lithuania) with an adjustable repetition rate, pulse energy, and wavelength (532 or 1064 nm) was used. We used the 532nm excitation with a repetition rate of 200Hz. By utilizing a laser with a wavelength near the absorption edge, we were able to selectively excite the material. The chosen repetition rate further enabled the observation of the slow recombination process characteristic of low-temperature conditions. We estimated the beam diameter size by optical inspection and we also varied it using conventional geometric optical elements. For the 200 Hz repetition, about 500 $\mu J/{\mathrm{cm}}^{2}$ corresponds to an averaged power density of 100 mW/${\mathrm{cm}}^{2}$, which is about 1 sun irradiance (1 kW/$m^{2}$).

The microwave reflectometry setup is illustrated in Figure 1. The microwave source (MKU LO 8-13 PLL, Kühne GmbH, Berg, Germany) was operated at 10GHz. A hybrid coupler (Micronde R433721, Radiall, Tempe, AZ, USA) was used to split the signal into two paths, with one providing the LO signal for the IQ mixer. To eliminate the DC reflection from the sample and isolate the AC component due to photoconductivity, a magic tee with a reference arm equipped with phase and amplitude tuning was employed. This configuration prevents mixer saturation and maintains the oscilloscope at maximum digital resolution. Isolators were placed to protect the microwave source from any returning signals, and DC blocks were added before the mixer. The IQ0618LXP mixer (Marki Microwaves, Morgan Hill, CA, USA) was configured with the reflected signal from the sample connected to the RF arm and the reference signal to the LO arm. A low-noise amplifier (JaniLab Inc., Kunszentmiklós, Hungary) was positioned between the magic tee and the IQ mixer.

Coplanar waveguides (CPWs) are planar structures consisting of a ceramic substrate with conducting areas separated by gaps, making them ideal for microwave applications. The coplanar waveguide used in this study is a conductor-backed coplanar waveguide (CBCPW), where the signal propagates through a central strip, while the two adjacent strips and the backside of the CPW serve as the ground plane. The front and backside ground surfaces are interconnected by via-holes through the ceramic layer, which are coated with the same conductive material as the surfaces.

The magnetic and electric fields of a CPW [47] are particularly advantageous for the measurements conducted in this study. The magnetic field is strongest and most uniform at the gap between the signal and ground strips on the front side, which leads to high signal levels when the sample is positioned at this location. By placing the sample on the CPW, mounted on the cold finger of the cryostat, the setup allows the CPW to function both as an antenna and as a cooling pad for the sample. With this setup, we directly measure the microwave signal reflected by the sample. Due to the DC signal reduction realized by the reference arm of the magic tee, we can measure the effect of excitation without the DC reflection coming from the sample. Using a phase shifter, placed between the source and the LO arm of the mixer, the phase difference between the RF and LO arm is also tunable thus providing a way to change the ratio of signal between the I and Q arm of the output.

### 2.3. Method of Detection

Our experimental setup measures the reflected microwave voltage in both in-phase and out-of-phase components due to the IQ mixer-based phase-sensitive detection. This measurement inevitably includes some residual DC background. The exciting, $U_{\mathrm{exciting}}$, and reflected, $U_{\mathrm{reflected}}$, voltages are related by the reflection constant, Γ, also known as the $S_{11}$ parameter, which is generally constant, though there may be a phase shift in the reflected voltage. Reflection occurs as the presence of the sample on the CPW locally perturbs its wave impedance, $Z_{0}=50\;\Omega$ [48], leading to $Z_{\mathrm{perturbed}}$. This results in the well-known relationship between the reflection coefficient and the perturbed impedance:(1)$$\Gamma =\frac{U_{\mathrm{reflected}}}{U_{\mathrm{exciting}}}=\frac{Z_{\mathrm{perturbed}}-Z_{0}}{Z_{\mathrm{perturbed}}+Z_{0}}.$$

The perturbation occurs via the so-called surface impedance of the sample, $Z_{s}$. The concept of surface impedance is a convenient way to describe the high-frequency properties (radiofrequency or microwave) of a material, but mathematically, it is equivalent to using the concept of the complex refractive index in optics. Its explicit form at an angular frequency of ω reads:(2)$$Z_{s}=\sqrt{\frac{i\omega \mu }{\sigma +i\omega \epsilon }},$$
which contains magnetic (through the permeability, μ), dielectric (through the permittivity, ϵ), and conducting (through σ) effects.

In the presence of a sample, the resulting perturbed CPW impedance is given by: $Z_{\mathrm{perturbed}}=\eta Z_{s}$. Here, the η filling-factor depends on the sample size and describes how the sample covers the gap between the grounding and the center conductor of the CPW. Its exact value is sample-dependent and remains unknown (but also irrelevant) in our study.

Microwave photoconductivity measurements often assume [49] that the reflected voltage and the sample conductivity are linear for a reasonable σ range. This can be derived by using Taylor series expansion of the surface impedance and the reflection coefficient for small conductivity changes. We obtain: (3)$$Z_{s}\left(\sigma +\Delta \sigma \right)\approx Z_{s}\left(\sigma \right)\cdot \left(1-\frac{\Delta \sigma }{2(i\omega \epsilon +\sigma )}\right)$$

Leading to:(4)$$\Gamma \left(\sigma +\Delta \sigma \right)\approx \Gamma_{0}\left(1-\frac{Z_{0}\cdot Z_{s}\left(\sigma \right)}{\left(Z_{s}{\left(\sigma \right)}^{2}-Z_{0}^{2}\right)}\cdot \frac{\Delta \sigma }{i\omega \epsilon +\sigma }\right)$$

The final result is that the additional microwave reflection due to sample, ΔΓ, reads: (5)ΔΓ∝Δσ∝Δn
where Δn is the excess charge-carrier concentration, dark charge-carrier concentration, reflectivity coefficient, and conductivity for the sample without light excitation. The above equation also means that the relationship between the measured reflected microwave voltage, $U_{\mathrm{reflected}}$ (denoted as *U* for simplicity in the following), and the excess charge-carrier concentration Δn is linear, i.e., U=C·Δn.

The penetration depth of the electromagnetic radiation is $\delta =\sqrt{\frac{2}{\mu \omega \sigma }}$. In our case, ω=2π·10GHz, and as the sample is non-magnetic, the vacuum permeability can be used. According to literature values [50] the dark resistivity is around 0.1GΩcm, giving a dark penetration depth of δ=5m. This is assumed to be reduced to 10Ωcm upon exposure to light, giving δ=1.6mm, which is smaller or comparable to the sample thickness. This means that we probe the conductivity in the bulk of the sample.

### 2.4. Analysis of the Data

We measure the magnitude of the reflected microwave radiation, U(t), as a function of time, following a pulse. From this quantity we then evaluate the apparent charge-carrier recombination time [49,51], $\tau_{c}$, using the rate equation with *n* being the excess charge carrier concentration: (6)$$\frac{\partial n(t)}{\partial t}=-\frac{n}{\tau_{c}}$$
which after rearrangement gives:(7)$$\tau_{c}=-\frac{n(t)}{\frac{\partial n(t)}{\partial t}}.$$

Using the above-mentioned U(t)∝n(t) assumption, it leads to the final result used in our analysis:(8)$$\tau_{c}=-\frac{U(t)}{\frac{\partial U(t)}{\partial t}}.$$

Here, the derivative is obtained by a direct numerical derivation from the data. The advantage of this approach is that it allows for the analysis of decay curves where the recombination time changes along the trace itself. We note that this approach also assumes that the charge-carrier mobility does not change during a photoconductivity decay trace. This could only happen if the charge-carrier concentration is high enough to result in significant carrier-carrier interaction effects, which is not reached in our case. Our sample has a volume of about $V=0.004\;{\mathrm{cm}}^{3}$, which absorbs a pulse of 70 μJ (this is our typical pulse energy, not to be mistaken with the pulse energy density on the sample). Assuming 100% quantum efficiency for the charge-carrier excitation, this gives rise to an excess charge carrier concentration of about $\Delta n\approx 4\cdot 10^{16}\;1/{\mathrm{cm}}^{3}$, which is a rather moderate value where no significant carrier-carrier interactions are expected. In addition, the time-resolved detection of photoconductivity has the inherent advantage that its determination of $\tau_{c}$ is not influenced by temperature-dependent changes of the mobility, which in turn is present in lead-halide perovskites [52]. On the other hand, continuous-wave or quasi-steady state photoconductivity studies are influenced by the temperature dependence of the mobility [53,54], and thus, a careful calibration of the temperature-dependent conductivity is required prior to the PCD studies [55].

Given the high density of time points and the noisy nature of the numerical derivative, we apply an adaptive smoothing to the data. First, we determine the maximum value from the initial time points following a pulse and the minimum value by averaging the noise long after the pulse. The data are then segmented into *N* logarithmically distributed bins (typically *N* = 50–100) between these minimum and maximum values, reflecting the exponential decay of the time traces. The data points in each bin are averaged into a single value, effectively smoothing the noisy data adaptively.

The logarithmic binning results in finer segmentation at the beginning of the decay, where the signal-to-noise ratio is higher and recombination is faster, and coarser segmentation at the end, where the process slows and noise reduction is more critical. We have typical data traces when the initial relaxation is analyzed with a 60ns bin size, while the end of the traces uses an 80μs-long bin. The effectiveness of the logarithmic binning method is demonstrated in Figure 2. This method balances time resolution and noise reduction, but careful selection of bin size is crucial. Smaller bins may lose fast recombination components, while larger bins may obscure or distort slow recombination near the end of the decay. The binning method and a deep averaging of the data allows to measure a recombination time domain of three–four orders of magnitude (i.e., our measurement has a high dynamic reserve). These dynamics ranged from the order of 100 nanoseconds in the initial phase of recombination to milliseconds at the end of the decay process.

## 3. Results

Figure 3 presents the charge-carrier lifetimes from time-resolved microwave-detected photoconductivity decay (TRMCD) results for CsPbBr_3_ at several representative temperatures. To obtain these data, we applied the numerical differentiation technique described in the Section 2. The measurements were conducted under an average irradiance near one sun ($100\;\mathrm{mW}/{\mathrm{cm}}^{2}$). TRMCD, a non-contact and non-destructive technique, is widely employed in the semiconductor industry to assess the purity and quality of silicon wafers during various stages of production. As described in the Section 2, the TRMCD signal is directly proportional to the charge-carrier density, and its time dynamics are primarily governed by charge recombination processes, unaffected by the temperature-dependent mobility. This method provides insights into the photogenerated charge-carrier density, mobility, and the recombination time, $\tau_{c}$, of the photoexcited carriers [53,54,56]. The value of $\tau_{c}$ is critical in determining whether the generated carriers can reach the solar cell edges, significantly impacting the photovoltaic efficiency.

Figure 3 presents the $\tau_{c}$ data as a function of the reflected microwave voltage, which is proportional to Δn under reasonable assumptions, as discussed in the Section 2. The behavior of the $\tau_{c}$ curves is unexpected. In silicon, all TRMCD curves for different laser energies converge into a single curve [49]. However, for CsPbBr_3_, the curves shift progressively toward higher reflected microwave voltages, with minimal changes in shape. As we demonstrate below, this behavior is characteristic of a trap-dominated charge-carrier recombination mechanism. There is increasing evidence of charge-carrier trapping in metal-halide perovskites, both organic [57] and inorganic [58], and a theoretical framework for this effect was outlined in Ref. [59].

The most important observation is that the voltage (or charge-carrier level)-dependent $\tau_{c}$ curves do not fall on one another and these strongly depend on the initial injection level. At low temperature, the curves start at around $\tau_{c}=10\;\mu s$ and saturate around 1 ms. The change in $\tau_{c}$ is thus about three orders of magnitude. Another important observation is that during the rapid charge-decay regime, the TRMCD signal itself drops about by a factor of 2, which is evident from the horizontal scale in Figure 3a. As we discuss below, it also supports that one type of charge carriers rapidly vanishes as these are selectively trapped.

At room temperature, the starting value immediately after the pulse is $\tau_{c}=0.1\;\mu s$. The shape of the injection dependent curves also changes with temperature: at low temperature, the saturated $\tau_{c}$ behavior is replaced by a more gradual change between short and long $\tau_{c}$ values. However, the charge-carrier concentration dependent nature of the $\tau_{c}$ curves is retained in the whole temperature range. We demonstrate in the following the utility of injection-dependent studies in identifying the recombination mechanism.

## 4. Discussion

Following a light pulse, the out-of-equilibrium state is formed in a semiconductor with excess charge carriers. The decay processes, with characteristic recombination time $\tau_{c}$ are conventionally described by separating contributions from the bulk and from the surface. Given that these are parallel processes, the contribution rates to the recombination are additive [60]: (9)$$\frac{1}{\tau_{c}}=\frac{1}{\tau_{c,\mathrm{surface}}}+\frac{1}{\tau_{c,\mathrm{bulk}}}$$

As mentioned in the Section 2, recombination in sizeable perovskite samples is dominated by the bulk processes. However, the presence of surface recombination cannot be fully ruled out but it is probably more relevant in thin films. The bulk contribution is conventionally split into three different recombination mechanisms, the radiative, the Auger, and the Shockley–Read–Hall (SRH) process. Again, the respective contributions to the recombination rates are additive:(10)$$\frac{1}{\tau_{c,\mathrm{bulk}}}=\frac{1}{\tau_{c,\mathrm{SRH}}}+\frac{1}{\tau_{c,\mathrm{radiative}}}+\frac{1}{\tau_{c,\mathrm{Auger}}}$$

In Figure 4, we present the band schematics of the band models and the underlying charge-carrier recombination mechanisms [61,62,63]. In the following, we present model calculations which are intended to shed light on the particular details of the experimental data. Our focus is on a simply attainable numerical modelling and qualitative understanding of the various recombination processes. We also neglect the residual (or dark) charge carrier concentrations in the lead-halide perovskites, and thus, it is assumed to be zero. The situation when this is not the case are discussed in the literature [60,64]. We justify this approach with two particular properties of the lead-halide perovskites: first of all, they have larger bandgap (above 2 eV [4,65]) than for most conventional photovoltaic semiconductors (e.g., for Si, the bandgap is 1.12 eV). This means that at room temperature the intrinsic or thermally excited charge-carrier concentration, $n_{i}$, is $2\cdot 10^{7}$ times smaller using the well known $n_{i}\propto \exp\left(-\frac{E_{g}}{k_{B}T}\right)$. Second, doping of the lead-halide perovskites in either *n* or *p* doping direction is less developed than for Si.

With this simplification, the rate equations corresponding to Figure 4 are readily obtained. The radiative recombination is characterized by a direct band-band recombination of an electron with a hole and the corresponding rate equation for the conduction band charge-carrier density (electrons), *n*, and the valence band charge-carrier density (holes), *p*, reads:(11)$$\begin{array}{cc} \frac{dn}{dt}= & -B_{\mathrm{rad}}\;n\cdot p\\ \frac{dp}{dt}= & -B_{\mathrm{rad}}\;n\cdot p. \end{array}$$
where the n·p product reflects that this is a two-particle process. This shows that the effective recombination time depends on the actual charge-carrier concentration as: $\tau_{c}=1/B_{\mathrm{rad}}p$ (the same expression is obtained using *n*). A similar dependence is obtained on the initial charge-injection value.

The extra energy following the recombination is taken away by an electron or hole in the Auger process. It is, thus, a three-particle process and the corresponding rate equations are:(12)$$\begin{array}{cc} \frac{dn}{dt}= & -C_{h,\mathrm{Auger}}\;n^{2}\cdot p-C_{e,\mathrm{Auger}}\;n\cdot p^{2}\\ \frac{dp}{dt}= & -C_{h,\mathrm{Auger}}\;n^{2}\cdot p-C_{e,\mathrm{Auger}}\;n\cdot p^{2}, \end{array}$$

The rate equations show that the corresponding lifetimes depend even stronger on the charge-carrier concentration for the Auger process than for the radiative one. The Auger recombination therefore limits the lifetime for very large charge-carrier concentrations.

Figure 5 shows charge-carrier lifetimes from a solutions of the above two sets of rate equations for the radiative (Equation (Equation 11)) and Auger processes (Equation (Equation 12)), as well as for a combined process. We used a simple model with parameters including the recombination rates, $B_{\mathrm{rad}}$, $C_{\mathrm{Auger}}$ and injection levels, n(t=0) given in the figure. Clearly, the Auger process dominates for high injection levels whose contribution rapidly vanishes during the recombination process. It is also important to note for the discussion below that all $\tau_{c}$ data fall on the same curve irrespective of the starting injection level.

The Shockley–Read–Hall process [61,62] is also known as the trap-assisted process. The most widely used (and simplest) model considers a deep-level trap state (a level which is further away from both the conduction and valence bands than $k_{B}T$). The model considers that the SRH trap state can capture an electron with probability $C_{e,\mathrm{SRH}}$, it can emit an electron (if it is occupied by one) with probability $E_{e,\mathrm{SRH}}$, a hole can be captured in the SRH state with probability $C_{h,\mathrm{SRH}}$ (this also corresponds to the emission of an electron from the trap into the valence band), and a hole can be emitted from the SRH trap state into the valence band with probability $E_{h,\mathrm{SRH}}$. The last process corresponds to the thermal activation of an electron from the valence band into the SRH trap state.

In the model, the density of SRH trap states is denoted by $N_{\mathrm{SRH}}$ and their occupational density with an electron by $n_{\mathrm{SRH}}$. For this three level system, we readily obtain the corresponding rate equations as:(13)$$\begin{array}{cc} \frac{dn}{dt}= & -C_{e,\mathrm{SRH}}\left(N_{\mathrm{SRH}}-n_{\mathrm{SRH}}\right)n+E_{e,\mathrm{SRH}}n_{\mathrm{SRH}},\\ \frac{dn_{\mathrm{SRH}}}{dt}= & +C_{e,\mathrm{SRH}}\left(N_{\mathrm{SRH}}-n_{\mathrm{SRH}}\right)n-E_{e,\mathrm{SRH}}n_{\mathrm{SRH}}\\ & -C_{h,\mathrm{SRH}}n_{\mathrm{SRH}}p+E_{h,\mathrm{SRH}}\left(N_{\mathrm{SRH}}-n_{\mathrm{SRH}}\right),\\ \frac{dp}{dt}= & -C_{h,\mathrm{SRH}}n_{\mathrm{SRH}}p+E_{h,\mathrm{SRH}}\left(N_{\mathrm{SRH}}-n_{\mathrm{SRH}}\right), \end{array}$$

Note that the *C* and *E* coefficients have differing physical dimensions. Charge neutrality dictates that $p=n+n_{\mathrm{SRH}}$, which is clearly satisfied by Equation (Equation 13). We note that Equation (Equation 13) assumes a trap state which can be negatively charged. As the problem of positive and negative charge states is fully symmetric, this can be complemented with a positively charged trap state. The corresponding set of equations is given in the Appendix A.

The conventional SRH result is an injection level independent $\tau_{c}$, whose magnitude is inversely proportional to $N_{\mathrm{SRH}}$. We note that for doped semiconductors a distinction is made between low- and high-injection levels for the SRH process depending on the injection level magnitude with respect to the equilibrium charge-carrier concentrations [60]. This distinction is, however, not relevant in our case. The SRH process dominates the recombination in semiconductors for injection levels, when radiative and Auger contributions can be neglected, which makes the TRPCD method a useful tool to characterize semiconductor purity [60].

The conventional SRH situation is recovered from Equation (Equation 13). by setting $C_{h,\mathrm{SRH}}=C_{e,\mathrm{SRH}}$ and $N_{\mathrm{SRH}}<n\left(0\right)$ and also that the emission rates are zero. This scenario is equivalent to assuming a deep-level (or mid-gap) state as then the thermally activated excitations (which correspond to the emission processes) can be neglected especially at low temperatures. The last condition describes that the number of available trap states constitutes a bottleneck, i.e., after a short while the trap states are always occupied. A straightforward consideration gives then that $\tau_{c}=\frac{2}{N_{\mathrm{SRH}}C_{e,\mathrm{SRH}}}$ and a simulated result is an injection-dependent horizontal line (shown in the Appendix A).

This conventional SRH result is clearly not capable of explaining the experimental data, which demonstrates a strong injection dependence of the $\tau_{c}$ curves. Especially pronounced is the rapidly changing $\tau_{c}$ immediately after the exciting pulse, which dominates the low-temperature behavior.

A numerical analysis (the computer code is provided in the Appendix A) of the solutions to Equation (Equation 13) reveals that the rapid change in $\tau_{c}$ can only be explained by a significant asymmetry between $C_{e,\mathrm{SRH}}$ and $C_{h,\mathrm{SRH}}$ (while assuming the corresponding emission rate to be zero). In fact, the ratio $\frac{C_{e,\mathrm{SRH}}}{C_{h,\mathrm{SRH}}}$ fixes the magnitude of $\tau_{c}$ change after the pulse. At the lowest temperature, this ratio is about 100 as $\tau_{c}$ changes from 10 μs to 1 ms. We also observed numerically that for the starting $\tau_{c}$ (immediately after the pulse), the above established $\tau_{c}=\frac{2}{N_{\mathrm{SRH}}C_{e,\mathrm{SRH}}}$ relationship holds.

Another constraint is that $N_{\mathrm{SRH}}$ has to be smaller than the studied injection levels or otherwise no flat dependence of $\tau_{c}$ on the charge-carrier concentration is observed. In our model studies, we set the maximum studied injection level to be n(t=0)=1, and thus, $N_{\mathrm{SRH}}$ is measured relative to this. As a result of all the above considerations, we find that $C_{e,\mathrm{SRH}}=10^{8}$, $C_{h,\mathrm{SRH}}=10^{6}$, and $N_{\mathrm{SRH}}=0.002$ explains well the low temperature curves and the result in shown in Figure 6. We underline that surprisingly our model essentially includes two parameters and both are fixed by experiment: the ratio of the electron and hole-capture rate is set by the minimal and maximal $\tau_{c}$ and in this model $N_{\mathrm{SRH}}$ behaves as a vertical scaling parameter.

We also discuss whether rapid exciton formation could account for the experimental observation. Excitons are electron-hole pairs which are bound by the Coulomb interaction [66] which are characterized by a finite binding energy with respect to the band-band separation of the corresponding uncorrelated electron-hole pair. It was found that exciton may indeed form in CsPbBr_3_ with a binding energy of around 40 meV [4,67], which means that these levels could in principle be rapidly populated at 20 K while being thermally stable. While we cannot entirely rule out the presence of bound excitons being responsible for the observed rapid vanishing of the charge carriers following a pulse, we argue that the trapping of one type of charge carriers is a lot more probable explanation. When exciton formation is present, one would expect that all charge carriers form this level following the pulse. However, in our experiment we observe that about half of the free charge carriers remain following the initial rapid change of $\tau_{c}$.

To simulate the high-temperature behavior with a minimal model, we first remind ourselves that at high temperatures a larger number of trap levels is expected as thermal excitation may ionize atoms which provide the trap centers. This means that the $N_{\mathrm{SRH}}$ parameter is expected to be significantly increased. We note that the ionization process actually increases the number of the available trap states, not their absolute number as the latter is built into the crystal. The probability of ionization for a level embedded in a semiconductor is [68]: (14)$$\begin{array}{c} P \end{array}\left(\mathrm{SRH}\;\mathrm{ionized}\right)=\frac{1}{1+2\cdot \exp^{\frac{\Delta E}{k_{B}T}}},$$
where the factor two is due to the spin degeneracy of electrons on the trap levels, ΔE is the energy difference between the chemical potential μ (same as Fermi energy at zero temperature), and $\varepsilon_{\mathrm{SRH}}$ is the energy of the SRH level. Given that at 20 K the thermal energy is 1.7 meV and at room temperature it is 26 meV, even a moderate ΔE=20 meV gives that at room temperature $3\times 10^{4}$ more trap levels can be ionized than at low temperature. These trap levels can thus become active for the SRH process.

Besides an increased $N_{\mathrm{SRH}}$, our numerical analysis showed that the experimental observations can only be explained if we simultaneously increase the hole-capture rate. A physical possibility is that overlap between the trap levels and the valence band increases. Alternatively, the thermal excitation of holes may lead to their increased kinetic energy which increases the so-called thermal velocity, leading to a higher hole-capture rate [60,69,70].

In Figure 6, we show the numerically obtained $\tau_{c}$ curves as a function of the excess charge-carrier concentration for the same initial injection levels that were used to simulate the low temperature behavior. Two parameters were changed: $N_{\mathrm{SRH}}$ was increased by a factor of 100 compared to the low-temperature simulation and $C_{h,\mathrm{SRH}}$ was increased by a factor of 20. As mentioned at the experimental results, the starting $\tau_{c}$ shortens by about a factor of 100 as compared to the low temperature ($\tau_{c}=10\;\mu s$ changes to about $\tau_{c}=0.1\;\mu s$), which essentially fixes the value of $N_{\mathrm{SRH}}$. We note that the lower set of curves in Figure 6 reproduces the experimental data at higher temperatures in Figure 3.

In Figure 7, we show the charge-carrier lifetime as a function of the detected voltage following a moderate exciting pulse energy density of 170 $\mu J/{\mathrm{cm}}^{2}$ for some selected temperatures. This excitation corresponds to about one third of a 1 sun irradiance. The data are normalized to the same starting voltage value as changes in the sample photoconductivity gives rise to changing voltage values. The displayed features are essentially the same as shown previously in Figure 3, i.e., that at low temperatures a rapid change in $\tau_{c}$ is observed, whereas at high temperatures, the $\tau_{c}$ vs. voltage change is more gradual.

Figure 7 also shows the results of numerical modelling using Equation (Equation 13) (the numerical code is provided in the Appendix A). We used the same parameters as in Figure 6: a fixed $C_{e,\mathrm{SRH}}=10^{8}$ and a logarithmically varying $C_{h,\mathrm{SRH}}$ (between $10^{6}$ and $10^{8}$) and $N_{\mathrm{SRH}}$ (between 0.002 and 0.2). An initial injection rate of n(t=0)=0.1 was used and again the horizontal axis is normalized to 1. We find that the major features of the experiment is well captured in the modeling results, the change of the above two parameters properly described the changing character of the $\tau_{c}$ vs. voltage curves, as already expected from the comparison of Figure 3 and Figure 6, but it is better demonstrated herein for a number of different temperature data. This reinforces our view that our model appropriately describes the observed behavior with a remarkably few free parameters and their physically supported change as a function of temperature.

## 5. Conclusions

In summary, we performed temperature- and power-dependent time-resolved microwave photoconductivity decay (TRMCD) measurements on CsPbBr_3_ single crystals in the temperature range of 20–300 K around one sun irradiance. Two aspects are novel in these studies: first, we systematically followed the changing charge-carrier lifetime after the exciting pulses to yield information about the recombination mechanisms. Second, we studied in detail the variation of these curves as a function of varying the initial charge-carrier injection. We argue that this approach is necessary to shed light on the recombination mechanisms in semiconductors in general. We also find that it is a sufficient approach to describe the mechanism when complemented with a modeling using the rate equations for the relevant bands. We observe from the detailed modeling that the presence of a charge-carrier trap state dominates the charge-carrier lifetime. Variation of a few physical parameters allows to qualitatively explain the observed features for the whole temperature range. Charge-carrier trapping limits the mobility of one type of carrier, posing challenges for direct photovoltaic applications. However, at the same time the ultra-long lifetimes may be beneficial for alternative applications including photodetection, light emission, and quantum memory storage.

## Figures and Tables

**Figure 1 nanomaterials-14-01742-f001:**
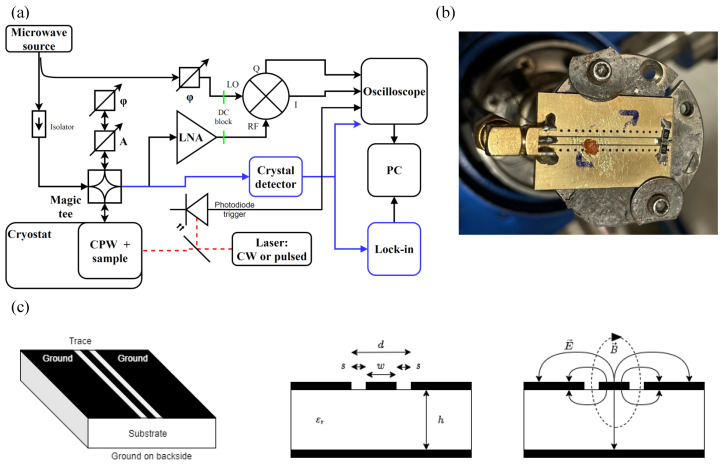
(**a**) Block diagram of the photoconductivity measurement system. It can be operated with CW and pulsed lasers as well and the signal may be recorded with a lock-in amplifier or an oscilloscope. The measurement can be realized with a low-noise amplifier and IQ mixer or a crystal detector (alternative setup with blue lines). (**b**) The CsPbBr_3_ sample is shown on a coplanar waveguide; note the pairs of 100Ω resistors which provide termination. (**c**) Schematics of the CPW and the corresponding electromagnetic field lines. The sample is placed over the gap of the CPW where the stray magnetic field is the strongest.

**Figure 2 nanomaterials-14-01742-f002:**
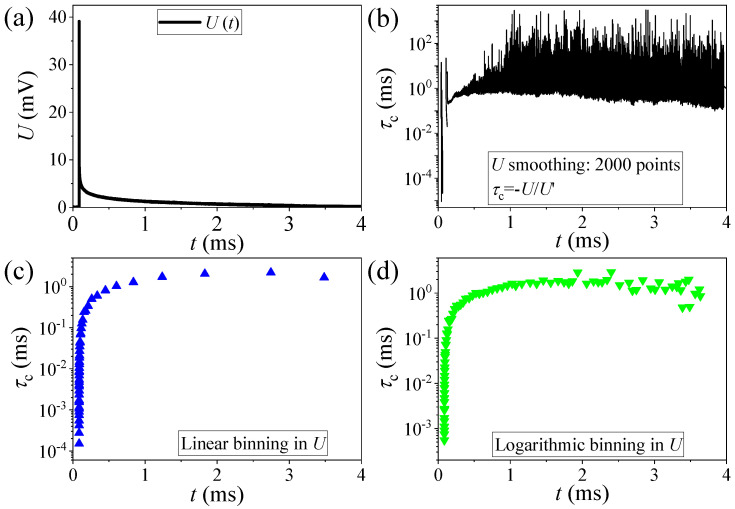
(**a**) Raw time-domain voltage signal with the averaged values of the binning methods shown on top of the signal. (**b**) Charge-carrier recombination dynamics calculated by smoothing the signal before the numerical derivation. (**c**,**d**) Charge-carrier recombination dynamics calculated by binning the measured data into linearly (**c**) or logarithmically distributed bins (**d**). Note the much more uniform distribution of data points with the latter method.

**Figure 3 nanomaterials-14-01742-f003:**
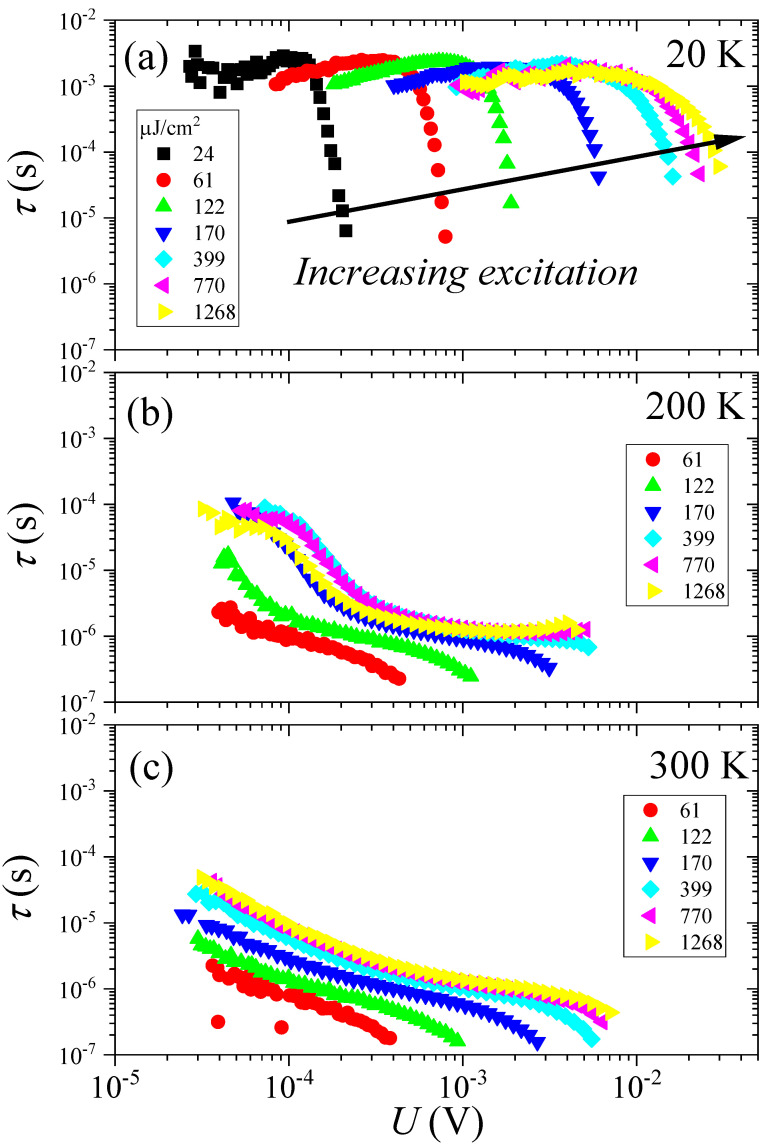
Injection level dependent charge-carrier lifetime values at different temperatures as a function of the detection voltage. The values are presented at 20K in subfigure (**a**), at 200K in subfigure (**b**), and at 300K in subfigure (**c**). Arrow indicates the direction of the increasing excitation energy which is the same as increasing injection level. Note that data for the smallest pulse energy density $24\;\mu J/{\mathrm{cm}}^{2}$ are not shown at higher temperatures.

**Figure 4 nanomaterials-14-01742-f004:**
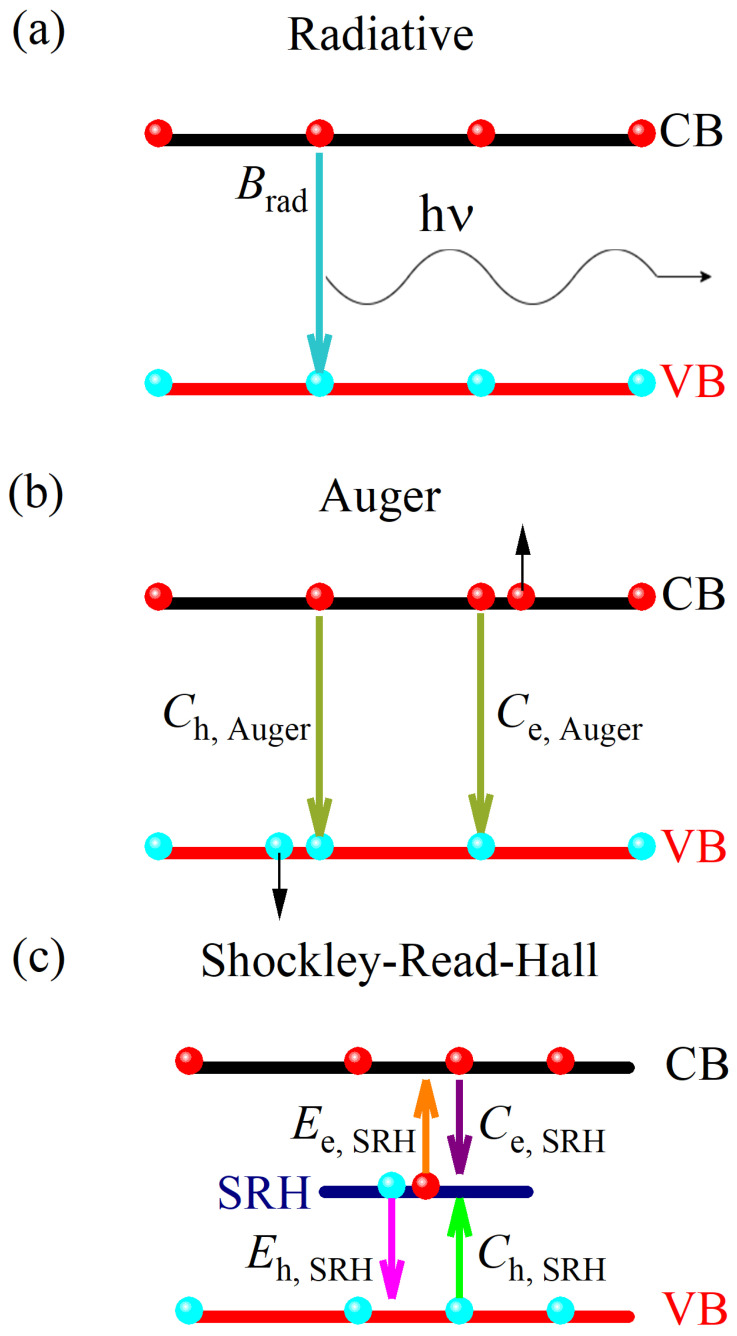
Simple band model of the most common charge carrier recombination mechanisms in semiconductors. Radiative-, Auger-, and trap-assisted or Shockley–Read–Hall (SRH) recombination mechanisms are presented in subfigures (**a**), (**b**), and (**c**), respectively. Lattice vibration (phonons) take away the energy typically for the SRH process, whereas an emitted photon and a kicked-out particle (an electron or a hole) takes away the energy for the radiative and Auger processes, respectively.

**Figure 5 nanomaterials-14-01742-f005:**
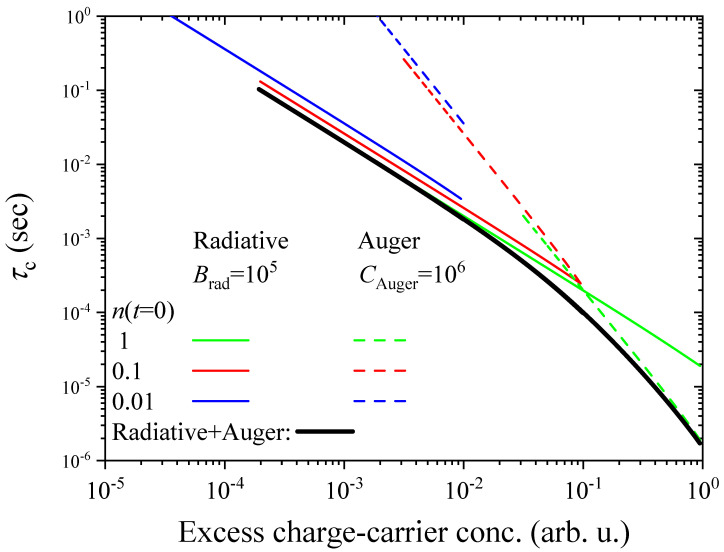
Charge-carrier recombination time as a function of the excess charge-carrier concentration for various initial injection levels (n(t=0)) and mechanism. The respective solid and dashed colored lines fall on the same curve but are offset for better visibility. The $C_{\mathrm{rad}}$ and $C_{\mathrm{Auger}}=C_{e,\mathrm{Auger}}=C_{h,\mathrm{Auger}}$ are given, but these have different units due to a differing definition. The $\tau_{c}$ result for the simultaneous presence of both processes is shown with a solid black line.

**Figure 6 nanomaterials-14-01742-f006:**
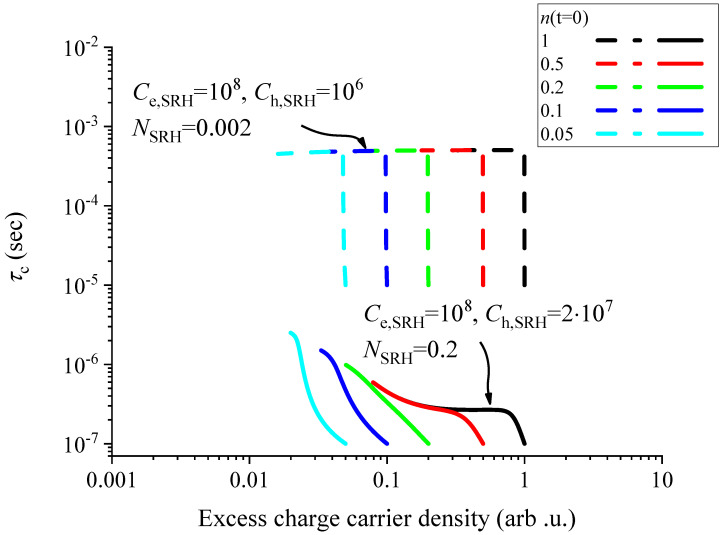
Charge-carrier recombination time as a function of the excess charge-carrier concentration for various initial injection levels, n(t=0), and emission capture rates. The upper lying set of curves reproduces well the low-temperature, whereas the lower curves reproduce well the higher temperature behaviors. The scale is the same as in Figure 4.

**Figure 7 nanomaterials-14-01742-f007:**
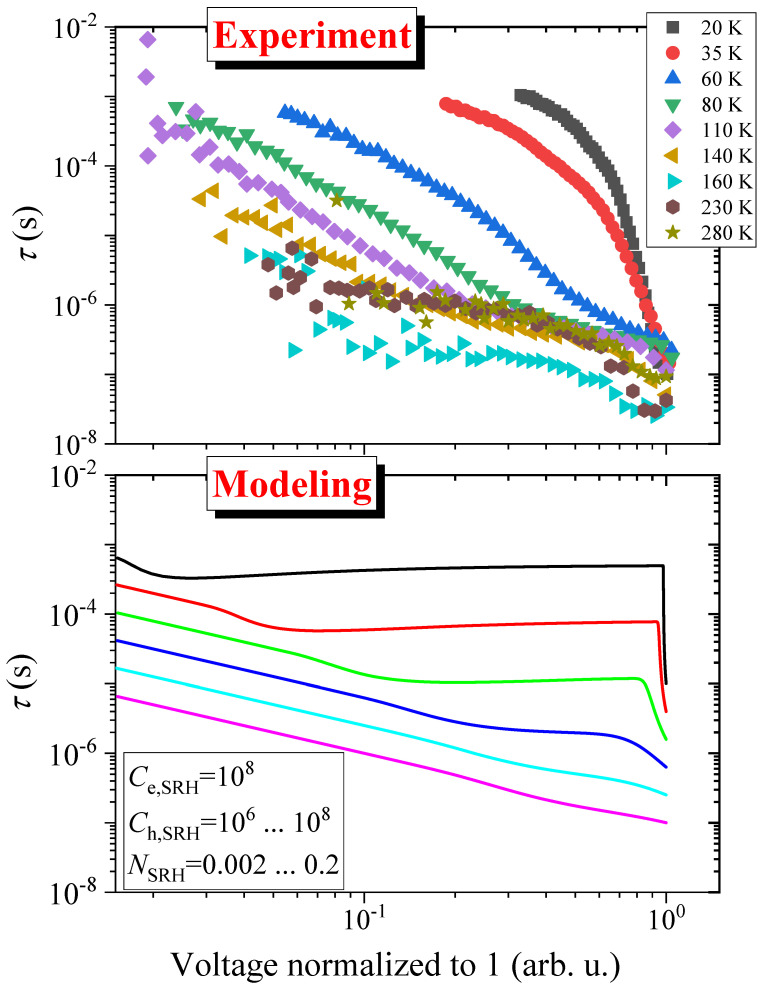
Temperature dependence of the charge-carrier lifetime at different temperatures for a fixed initial laser excitation energy of 170 $\mu J/{\mathrm{cm}}^{2}$. The voltage axes of all the data are normalized to the same value due to the temperature dependence of the photoconductivity of the sample. A simulated set of curves is also shown for a fixed electron-capture rate, $C_{e,\mathrm{SRH}}$ and a logarithmically varying $C_{h,\mathrm{SRH}}$ (between $10^{6}$ and $10^{8}$) and $N_{\mathrm{SRH}}$ (between 0.002 and 0.2). These are the same values as those show in Figure 6 for n(t=0)=0.1.

## Data Availability

The raw data supporting the conclusions of this article will be made available by the authors on request.

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
