# Peer review of "Dynamics of Photoinduced Charge Carriers in Metal-Halide Perovskites"

_nanomaterials, 2024, doi:10.3390/nano14211742_

Round 1
Reviewer 1 Report
Comments and Suggestions for Authors
In the manuscript by Bojtor and coworkers, the authors performed microwave photoconductivity measurements on lead halide perovskite CsPbBr3. By varying the photo-injection level from the optical pump and sample temperature, a significant change in the charge carrier recombination dynamics was observed. Instead of converging into the same decay time, the experimental data shifts progressively towards the initially higher injection level. This phenomenon becomes more obvious at lower temperatures. Through detailed analysis of different types of recombination channels including radiative, Auger and trapping recombination, the authors were able to distinguish individual contributions and eventually identify the key role of charge trapping in the ultralong carrier lifetime. Considering the fast development of the lead-halide-perovskite-based optoelectronic applications, it’s extremely relevant and important to investigate the photoexcited charge carrier lifetime. Due to the relatively limited decay time investigated by compensated spectroscopic tools like photoluminescence or transient absorption spectroscopy, microwave photoconductivity measurement could provide important information on much longer time scale of the charge carrier recombination process while maintaining good time resolution.
Overall, I would be happy to recommend this manuscript for publication in Nanomaterials, if the following questions about data interpretation can be addressed.
Comments:
1. The main observation in this manuscript is the photo-injection dependent lifetime curves as a function of the detected voltage shown in Figure 3, especially at low temperatures like 20 K. There is a fast increase of the charge carrier lifetime, i.e., fast initial decay in dynamics, for all the injection levels from 24 uJ/cm2 to 1268 uJ/cm2. The authors assigned it to the defect trapping. Yet, there is other possibilities that can account for such changes, for example, electron-hole pair (exciton) formation. The exciton binding energy for bulk CsPbBr3 has been reported to be around 40 meV (Nano Lett. 15, 3692–3696 (2015); Nat. Commu. 14, 1047 (2023)), which is much larger than the thermal energy at 20 K. How sensitive is the microwave photoconductivity measurements to the exciton species? Is it possible to distinguish the recombination dynamics from free charges and excitons from the experimental data?
2. The authors assume a linear relation between the detected voltage and the photoexcited carrier density in the data analysis for all injection levels. However, the charge mobility can be strongly affected by the carrier density due to carrier-carrier interaction. Considering the large variation of the pump power, it’s important to clarify the effect of the mobility change to the lifetime curves.
3. Comparing the different temperatures, the starting tc shortens by a factor of 100. The authors justify this effect in the model by varying NSRH. However, it’s well studied that the charge mobility can be changed dramatically with temperature in lead halide perovskites (Adv. Funct. Mater. 25, 6218-6227 (2015)). Since the calculation of tc is related to the detected voltage (equation 8) which is proportional to the product of the carrier density and charge mobility, did the authors consider the temperature-dependent mobility in the data analysis?
4. At high temperatures, the authors claimed an increased number of trap levels NSRH due to the thermal excitations. I assume the authors mean the available trap states, not the total trap states. The authors need to clarify better in the manuscript for the readers.
5. There is necessary reference missing in Page 7 for the discussion about the tc curves in silicon ‘In silicon, all TRMCD … into a single curve’.

Author Response
The reply is provided in the attached pdf file.

Reviewer 2 Report
Comments and Suggestions for Authors
In this work, the author measured temperature and power dependent time-resolved microwave photoconductivity decay (TRMCD) on CsPbBr3 single crystals at temperatures ranging from 20 to 300K in the presence of sun irradiation. In detail, the author analyzed the changes in carrier lifetime after excitation pulses and evaluated possible recombination mechanisms using mathematical models. In general, this work is well designed and the paper is well written. This paper maybe suitable for publication in nanomaterials after minor revision.
The detailed comments are as follows:
1. The writing of the Abstract section is not smooth enough, it is best to add some linking words.
2. Please fill in the keywords below the Abstract section.
3. In Chapter The Experimental Setup (2.2), “Figure 1 shows the block…” should be corrected to “Figure 1(a) shows the block…”. Each statement should be specific to each image. Please carefully check and make corrections.
4. Reference 53 contains a serious error, please correct it.
5. Image titles in the article should follow a consistent structure (Figure 1 in manuscript and FIG.1 in supporting information).
6. Whether title, author and other information should be added to the supporting information on the first page? Please standardize the format of your article according to the journal's requirements.
Comments on the Quality of English LanguageModerate editing of English language required.
Author Response
The reply is provided as an attached pdf file.

Round 2
Reviewer 1 Report
Comments and Suggestions for Authors
In the revised manuscript, Bojtor and coworkers carefully addressed all major and minor comments, their efforts are very much appreciated. With their addition of further discussions especially on mobility change and exciton formation, the conclusions are more convincing and sounder. The manuscript is now improved significantly. I recommend the manuscript for publication in Nanomaterials without another round of review. I congratulate the authors on this relevant and exciting piece of work.